# Trajectories of school absences across compulsory schooling and their impact on children's academic achievement: An analysis based on linked longitudinal survey and school administrative data

**Jascha Dräger[1], Markus Klein** [ID] **[2]\*, Edward M. Sosu[2]**

**1** German Institute for Economic Research, Mohrenstraße, Berlin (Mitte), Germany, **2** Strathclyde Institute of Education, University of Strathclyde, Glasgow, United Kingdom

\* markus.klein@strath.ac.uk

**Data Availability Statement:** The Millennium Cohort Study (MCS) sweeps used in the paper are publicly available at the UK Data Service (https://

## Abstract

Prior research has identified that school absences harm children's academic achievement. However, this literature is focused on brief periods or single school years and does not consistently account for the dynamic nature of absences across multiple school years. This study examined dynamic trajectories of children's authorised and unauthorised absences throughout their compulsory school career in England. It investigated the consequences of these absence trajectories for children's achievement at the end of compulsory schooling. We analyse linked administrative data on children's absences and achievement from the National Pupil Database and survey data from the Millennium Cohort Study for a representative sample of children born in 2000/2001 in England (N = 7218). We used k-means clustering for longitudinal data to identify joint authorised-unauthorised absence trajectories throughout compulsory schooling and a regression-with-residuals approach to examine the link between absence trajectories and achievement. We identified five distinct absence trajectories: (1) 'Consistently Low Absences', (2) 'Consistently Moderate Authorised Absences', (3) 'Moderately Increasing Unauthorised Absences', (4) 'Strongly Increasing Unauthorised Absences', and (5) 'Strongly Increasing Authorised Absences'. We found substantial differences between trajectory groups in GCSE achievement, even when accounting for significant risk factors of school absences. Compared to 'Consistently Low Absences', 'Strongly Increasing Unauthorised Absences' reduced achievement by -1.23 to -1.48 standard deviations, while 'Strongly Increasing Authorised Absences' reduced achievement by -0.72 to -1.00 SD for our continuous outcomes. 'Moderately Increasing Unauthorised Absences' (-0.61 to -0.70 SD) and 'Consistently Moderate Authorised Absences' (-0.13 to -0.21 SD) also negatively affected achievement compared to 'Consistently Low Absences'. Our research underscores the critical importance of examining entire trajectories of absenteeism and differentiating between types of absences to fully grasp their associations with academic outcomes and design targeted interventions accordingly.

ukdataservice.ac.uk/). "The Millennium Cohort Study: Linked Education Administrative Datasets (National Pupil Database), England: Secure Access" data (https://beta.ukdataservice.ac.uk/datacatalogue/studies/study?id=8481) used in the paper can be accessed via the UK Data Service's Secure Lab. It is the UK Data Service's flagship secure environment providing researcher's access to these sensitive and confidential data. Data accessed in this way cannot be downloaded. Once researchers and their projects are approved, they can analyse the data remotely from their organisational desktop, or by using their Safe Room. Access to the data for this project has been granted by the data providers and UK Data Service following an application process, including completion of project application form, ESRC research proposal, Secure Access User Agreement, ESRC Accredited Researcher Application, and Safe Researcher Training (SRT). For more information on how to apply to access con trolled data in the Secure Lab see: https://ukdataservice.ac.uk/find-data/access-conditions/secure-application-requirements/ or contact secure.applications@ukdataservice.ac.uk.

**Funding:** Our study is funded by the Nuffield Foundation (FR-000023241). The funders had no role in study design, data collection and analysis, decision to publish, or preparation of the manuscript.

**Competing interests:** The authors have declared that no competing interests exist.

## Introduction

Many countries worldwide have reported substantial increases in school absenteeism since schools reopened after the pandemic [1–4]. In turn, there is ample evidence of the harmful consequences of school absences on children's academic achievement [5–9], which later translates into lower educational attainment and poorer labour market outcomes [10, 11]. While this suggests that cumulative exposure to schooling over time determines a child's achievement [12], the school attendance literature has not comprehensively addressed the dynamic and multi-dimensional nature of absences and their effects on educational outcomes. Our article addresses three shortcomings in the literature.

First, school absences are not static but change as children progress through their education. The trajectory of absence may matter for children's achievement as it captures the extent, timing, and variability of exposure to school-based learning. However, the school absenteeism literature has not consistently examined the dynamic nature of school absences and their impact on achievement. Most studies measured absences in one year or averaged absences up to three school years [9, 13–15]. This restriction may mask important differences between students and likely underestimates the effect of absences on achievement [16]. The few studies examining absence trajectories and achievement [17–20] have primarily focused on primary and early secondary school stages. None of these studies has considered absence trajectories throughout a student's school career.

*Second*, studies have increasingly emphasised the importance of analysing the reason for school absence [21–24]. Absences can be caused by authorised (e.g., sickness) or unauthorised (e.g., truancy) reasons, which can change throughout a child's educational career and affect achievement differently. For instance, evidence from the UK context suggests that unauthorised absences are much more prevalent in later school stages [25]. Although all reasons for absences harm children's achievement [24], unauthorised absences are more detrimental to school performance than authorised absences [21, 22]. However, previous studies on absence trajectories and achievement have focused on absences as a whole [17, 20] and thus have yet to determine the intersecting role of absence reason and temporal variation of absences when analysing achievement. Examining the joint trajectories of authorised and unauthorised absences on achievement is crucial, given that they likely interact and vary in frequency over time.

Lastly, most of the existing studies on absence trajectories are based on regionally restricted samples in the US context [16, 17, 19, 26, 27], and there is limited research on the dynamics of absence trajectories from other contexts. This is significant because educational policies differ across countries, and we do not know yet whether there are more similarities or differences in the trajectories and consequences of school absences across countries. In addition, if studies link absence trajectories to achievement, they are unable to account for relevant baseline confounders of this association or fail to adequately account for time-varying confounders such as student behaviour, which may confound absences at a later stage but are influenced by absences at an earlier stage.

Our study addresses these gaps in the literature by identifying the joint trajectories of authorised and unauthorised absences throughout the compulsory school career using linked administrative data on absences and standardised achievement tests from the National Pupil Database (NPD) and survey data from the Millennium Cohort Study (MCS) for England. We use k-means clustering for longitudinal data (KML) to identify these typical school absence trajectories [28, 29]. In addition, we use a regression-with-residuals approach (RWR) and a rich set of confounders to analyse the extent to which the identified trajectories impact achievement at the end of compulsory schooling [30]. This novel approach allows us to

appropriately account for time-varying confounders, such as student behaviour, without introducing overcontrol and collider bias. We ask the following research questions:

1. What joint authorised-unauthorised absence trajectories emerge across compulsory school careers?

2. To what extent do these absence trajectories affect achievement?

## Absence trajectories and achievement

The Faucet theory suggests that students improve their skills through frequent exposure to schooling, and they cease making educational gains when their exposure to school is cut off [31]. Consequently, students who receive fewer hours of instruction during the school year are disadvantaged in their learning, receive lower grades, perform worse in exams, and are more likely to drop out of school [9]. This argument aligns well with empirical evidence demonstrating a link between classroom instruction time and academic achievement [32, 33]. In addition, students frequently absent from school may feel less connected to their classmates and struggle to participate in classroom activities and interactions with teachers and peers, which is detrimental to their academic development [34].

While there is abundant evidence on the negative consequences of school absences on children's school achievement [5, 7–9, 35], these studies did not consider that absences may be differently associated with achievement depending on their temporal sequencing and their reasons over time. The pattern of absences during primary and secondary school may vary over time and for different students across the school life span. Their detrimental effects may not manifest until after prolonged exposure [12]. While the literature on school absences focuses on measures of chronic absenteeism (typically 10% or more days) over a given school year, it does not account for the persistence of absences over multiple school years. For instance, absences from school in a single year may not be as disruptive to children's learning as absences in multiple years. Therefore, snapshot measures may underestimate the cumulative effect of school absences on later achievement. This requires a holistic measurement of school absences via clustering of individual trajectories.

Studies examining whether students vary in their absence trajectories during specific school stages (e.g., kindergarten to elementary; middle; or high school) have commonly found between four and seven clusters of absence trajectories for US students [18–20, 26]. However, only one study investigated the relationship between absence trajectories in elementary school and achievement [20]. Using data from the US Early Childhood Longitudinal Study–Kindergarten Class of 1998–1999 (ECLS-K), they identified four latent absence trajectories between kindergarten and fifth grade: a low absence trajectory (46%), a decreasing absence trajectory (24%), an increasing absence trajectory (22%) and a high absence trajectory (8%). Students in the low absence group performed the best in math and reading, while those in the high absence group performed the worst. No statistically significant difference existed between students with increasing and decreasing absence trajectories.

Absences may have varying consequences for achievement depending on the reason for absence. Teachers may view unauthorised absences negatively, resulting in increased student-teacher conflict, decreased closeness, and increased teacher irritation and frustration towards students who miss school without authorisation [36, 37]. As a result, teachers may be less willing to support students in catching up on missed lessons if they have taken unauthorised absences from school. Unauthorised absences are also linked to problem behaviours such as alcohol and substance abuse [38, 39] or crime and delinquency [40]. These behaviours may influence students' motivations to learn and can exacerbate the negative impact of absences on

achievement [41]. On the other hand, if students are absent from school for authorised reasons, they may be more motivated to make up for lost time and engage with missed lesson content. Teachers and parents may also be more willing to assist them in catching up on lesson content.

Some studies examining the impact of various reasons for absence have found that unauthorised absences were more harmful to achievement than authorised absences [5, 21, 22, 42]. A study on secondary pupils in Scotland found that excused absences due to sickness and exceptional domestic circumstances (e.g., bereavement) were just as damaging to achievement as unauthorised absences [24]. This may suggest that students' learning suffers if the excused reason prevents them from focusing on the missed lesson content. For example, authorised absences due to sickness may indicate health issues that negatively impact learning and achievement in the longer term [43].

Therefore, the temporal dynamics of authorised and unauthorised absences must be considered concurrently. However, previous studies have explored trajectories for overall absences [20, 26] or unauthorised absences such as truancy [19]. Only one study has examined the joint trajectory of authorised and unauthorised absences over a single high school year, assuming a single trajectory for all students [16]. They found an increase in unauthorised absences and a decrease in authorised absences across the school year. It remains to be seen whether multiple joint trajectories exist for distinct student groups across the school life span and the extent to which these trajectories are associated with achievement.

We advance this literature in several meaningful ways. First, we examine latent absence trajectories throughout students' entire school careers and study their impact on achievement at the end of compulsory schooling. Second, beyond the modelling of overall absences, we investigate profiles of joint trajectories of authorised and unauthorised absences and their consequences. Third, in contrast to previous research, our linked school administrative and survey data allow us to control for a comprehensive set of risk factors of school absences [44] and achievement, including time-varying confounders such as early cognitive ability, student behaviour and attitude towards school.

## Data and methods

### Data

For the analysis, we used data from the Millennium Cohort Study (MCS), a large-scale longitudinal study of children born in 2000 or 2001 and living in the UK (England, Northern Ireland, Scotland, and Wales) [45]. In total, 19,244 families (12,387 in England) were recruited and first surveyed when children were nine months old. Follow-up assessments took place at age 3 (sweep 2), age 5 (sweep 3), age 7 (sweep 4), age 11 (sweep 5), age 14 (sweep 6), and age 17 (sweep 7). These data are linked with the National Pupil Database (NPD), a register dataset of all students in state schools in England [46]. We received ethical approval for the study from the University of Strathclyde ethics committee in written form.

All MCS participants residing in England during sweeps 3–5 (N = 9,047) were asked for consent to link their data to the NPD. Consent was granted from 8,489 participants, and 8,438 were successfully matched to the NPD database. We restricted our analysis sample to participants who agreed to data linkage and were linked in sweep 4 (N = 8,206), which enabled us to use MCS survey weights. Moreover, we excluded all students for whom we lacked information on absences for a full academic year or achievement measures. This resulted in N = 7,218 cases for the analysis (see diagram on sample restrictions in S1 File).

We combined MCS weights and inverse probability weights [47] to correct for the non-random inclusion of students in our sample. Specifically, we multiplied MCS weights for participation in England in sweep 4 with the inverse of the probability that participants gave consent

to data linkage, have been successfully linked, and have complete absence and achievement data. We estimated the probability of being included in our analytic sample with a logistic regression using socio-demographic, child, and family characteristics as our predictors (see S2 File). By weighting the analysis this way, we created a pseudo-population with the same characteristics as the initial MCS sample of children living in England.

To address item non-response on control variables, we imputed missing values on covariates using multiple imputation based on Categorization and Regression Trees [CART, 48]. CART is a nonparametric recursive algorithm that creates groups with maximum intragroup and minimum intergroup homogeneity using binary splits. The advantage of using CART for multiple imputation is that the algorithm finds the best predictors of missing data from all potential covariates, including non-linear patterns and interactions. We created 20 imputed datasets and applied Rubin's rules to obtain standard errors [49].

## Variables

**School absences.**   School attendance policy in England is guided by the Education Act of 1996, which established the legal framework for school attendance. This legislation made parents or guardians legally responsible for ensuring their children's regular attendance at school from age five until the end of the academic year in which they turn sixteen. During our observation period, the UK government set school attendance targets and implemented truancy-reduction policies. Local education authorities and schools were responsible for monitoring attendance and taking appropriate action when attendance fell below acceptable levels. This includes issuing warnings, fines, or taking legal action against parents who do not ensure their children's regular attendance at school.

In the comprehensive school system of England, students attend primary school for six years (key stages 1 and 2) from ages 5/6 and compulsory secondary school for five years (key stages 3 and 4) from ages 11/12. Accordingly, the NPD contains information regarding the number of possible school days, the number of days missed due to authorised absences, and the number missed due to unauthorised absences in the autumn, spring, and first half of the summer term for each year of compulsory schooling, from year 1 (ages 5 to 6) to year 11 (ages 15 to 16). After the 2012/13 academic year, data on absences for the second half of the summer term were also collected [50]. To maintain consistency in measuring student absences over time and since we are interested in modelling absence trajectories across school years, we combined data from the fall, spring, and summer terms into annual absence data. Students could have attended about 1,700 school days in this period.

*Authorised absences* are those with permission from a teacher or other authorised school representatives, which is only granted if a satisfactory explanation for the absence, such as illness, has been provided. *Unauthorised absences* are absences for which the school has not granted permission. We calculated the percentage of days missed due to authorised or unauthorised absences because the number of possible days varies between years and between students within the same year. Average total, authorised, and unauthorised absences per year, correlations over time, and correlations between authorised and unauthorised absences are presented in the S3 File. Average authorised absences fall slightly from 4.9% in year one to 3.4% in year six, then rise to 4.5% in year eleven. On the other hand, unauthorised absences remain stable at around 0.5% in years 1–7 before rising to 2.0% in year 11. Over the school career, the association between authorised and unauthorised absences was small ($r = 0.08$ to $0.20$), indicating that these two types of absences are relatively distinct.

**Academic *a*chievement.**   The dependent variable is student achievement upon completion of compulsory schooling. Students sit for their GCSE (General Certificate of Secondary

Education) examinations at the end of year 11 (key stage 4), which are consequential for future education and labour market outcomes [51]. GCSE qualifications are offered in distinct subject areas. English and mathematics are considered "core subjects" that are compulsory. Although the precise number differs among individuals, it is customary for students to undertake a minimum of five GCSEs. A student is allocated a point value ranging from 1 (indicating the lowest) to 9 (representing the highest grade). For our analysis, we consider four different outcomes.

1. 'Attainment 8' score (Mean = 47.3, SD = 19.1, Range 0–90), combining scores from eight best-performing GCSE subjects, including English, math, and six additional exam results from GCSE-level qualifications. English and math are counted twice. Three GCSE subjects must come from qualifications that count towards the English Baccalaureate (EBacc), such as sciences, languages, or history;

2. GCSE Math attainment (Mean = 10.1, SD = 4.2, Range 0–18);

3. GCSE English attainment (Mean = 9.2, SD = 4.3, Range 0–18);

4. Whether students passed five or more GCSEs, including Math and English. Students passed a GCSE exam if they attained a point score of more than or equal to four. 59% of pupils in our sample passed five or more GCSEs.

For the analysis, we standardised the 'Attainment 8' score and GCSE English and math scores to have a mean of zero and a standard deviation of one in the weighted sample.

**Covariates.** The MCS enabled us to account for risk factors of school absenteeism identified in the literature [44], which also influence student achievement. We included multiple measures of students' socio-economic background (parental education, parental occupational class, household income, housing tenure, neighbourhood deprivation) and child and family demographics (gender, date of birth, ethnicity, family structure, household size, number of children, region). Moreover, we adjusted for students' and parents' attitudes towards school and educational aspirations, child behaviour problems (externalizing and internalizing), parental involvement in school, child health, birth conditions (e.g., birthweight, smoking during pregnancy), disruptive events in the student's life (e.g., residential mobility and school changes), whether the family pays tuition fees for the school, and which ability stream and set the child attends. Finally, we considered children's cognitive ability and performance in standardised achievement tests at the end of the second and sixth school years. We measured cognitive ability, educational motivation and aspirations, behaviour problems, parental involvement, ability streams and set, and school changes at multiple times points. A description of the timing of the measurement of covariates is shown in the S4 File, the measurement of latent covariates, such as attitudes towards school, is described in the S5 File, and the distribution of all covariates is shown in the S6 File.

## Methods

### Identifying absence trajectories

We applied k-means for longitudinal data (KML) to identify clusters of students with similar joint trajectories on authorised and unauthorised absences from years 1 to 11, using the R package kml3d [28, 29]. The optimal number of clusters is unknown and cannot be deduced from current research. Existing research has identified between four and seven distinct absence trajectory clusters based solely on total or unauthorised absences [19, 20, 26]. We tested cluster solutions with two to eight clusters and opted for an optimal cluster solution

based on fit indices, predictive validity, the number of observations in the smallest cluster, and interpretability. For more information on identifying the optimal cluster solution, see S7 File.

### Estimating effects of absence trajectories on achievement

To model the effect of absence trajectories on achievement, we accounted for important risk factors of school absenteeism [44], including time-varying covariates, such as early cognitive ability, educational motivation, and behavioural issues. However, some of these time-varying covariates may be both consequences of earlier absences and confounders for the effect of later absences on achievement [52]. Consequently, controlling for these time-varying confounders may remove some of the effects of earlier absences and result in overcontrol and collider bias. This issue is referred to as confounder feedback bias [47].

To address this problem, we employed a regression-with-residuals approach [RWR, 30]. In the first step, we regress each time-varying covariate on baseline covariates, all earlier measures of absences and all earlier measures of time-varying covariates, and obtain the residuals of these regressions. In the second step, we regress achievement on absence trajectories, baseline confounders, and the residualised time-varying confounders. The residualised risk factors that could be both the consequence of earlier absences and the cause of later absences are indicated by an "R" in S3 Table in S3 File (cognitive abilities, attitudes towards school, child behaviour, parental involvement, and school characteristics), while baseline risk factors which are not affected by absences are indicated by a "B" in S3 Table in S3 File.

We used linear regression models for Attainment 8, English and math attainment, and a linear probability model for obtaining five or more GCSEs. Under the assumption that there is no unmeasured confounding, positivity, and correct model specification, the RWR estimates for absence trajectories on our achievement outcomes can be interpreted as average causal effects in the English pupil population covered by the MCS. Given that our covariate set contains longitudinal and high-quality measures of risk factors for school absences, we believe these assumptions are plausible.

## Results

### Clusters of absence trajectories

Based on fit indices, predictive validity, and the number of observations in the smallest cluster, we identified five joint authorised-unauthorised absence trajectories. First, the five clusters resulted in the highest Calinksi-Harabasz index [29]. Second, it significantly increases the variance in achievement explained compared to two or three cluster solutions and nearly explains as much variance as seven or eight cluster solutions. Finally, the smallest cluster in the five-cluster solution is nearly twice the sample size of the smallest cluster in the six-cluster solution. The S7 File explains in detail why we prefer the solution with five clusters over other solutions. Fig 1 depicts the mean trajectory of authorised and unauthorised absences from Years 1 to 11 for the different clusters. The S8 File provides information on the exact percentages of absences each year and averaged across the 11 years for each identified cluster.

About two-thirds of students fall into a cluster characterised as a *Consistently Low Absence* (CLA) trajectory. On average, across 11 school years, students in this cluster had 2.6% authorised absences and 0.3% unauthorised absences (see S8 File).

The second largest cluster contains about 28% of students and is characterised by *Consistently Moderate Authorised Absences* (CMAA). In most years, authorised absences in this cluster ranged between 6% and 8%. Additionally, students had, on average, 1.0% unauthorised absences.

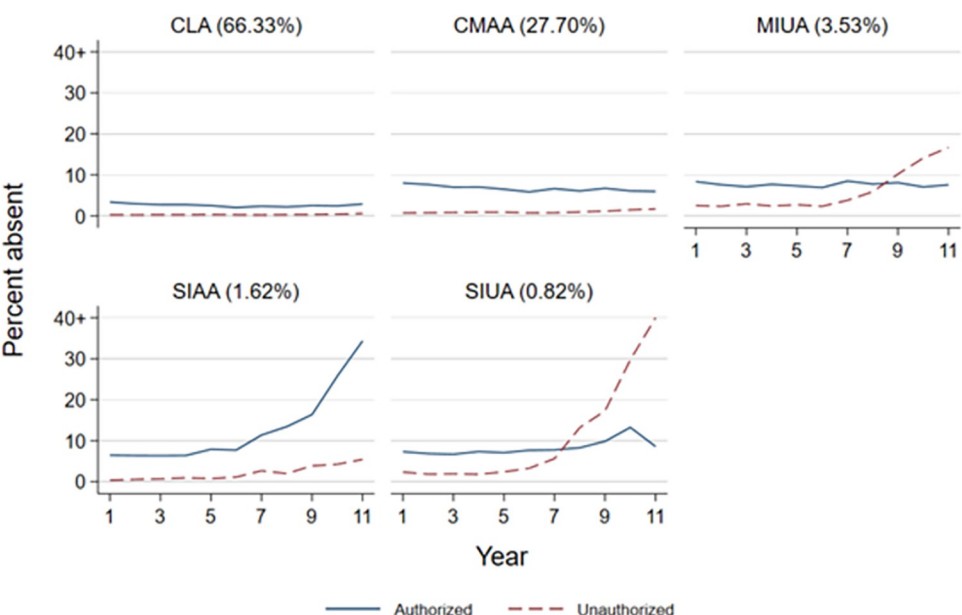

**Fig 1. Mean authorised and unauthorised absences over time by absence trajectory cluster.** *Note*. Linked MCS-NPD data, N = 7,218, weighted. CLA = Consistently Low Absence, CMAA = Consistently Moderate Authorised Absences, MIUA = Moderately Increasing Unauthorised Absences, SIAA = Strongly Increasing Authorised Absences, SIUA = Strongly Increasing Unauthorised Absences. Mean absences above 40% are truncated.

In addition to these large clusters, we found three smaller clusters of increasing absences. 3.5% of students belong to a cluster with *Moderately Increasing Unauthorised Absences* (MIUA). These students had comparatively high unauthorised absences in years 1 to 6, ranging between 2% and 3%, which increased to 17% in year 11. Moreover, students in this cluster consistently had 7% to 8% of authorised absences.

The penultimate cluster is characterised by *Strongly Increasing Authorised Absences* (SIAA), including 1.62% of the student sample. Authorised absences in this cluster increased from around 7% in primary school to more than 30% in year 11. Additionally, unauthorised absences increased in secondary school but remained below 6% even in the final years.

Students in the final cluster, *Strongly Increasing Unauthorised Absences* (SIUA), show moderately low unauthorised absences in primary school but have a much stronger increase in unauthorised absences in secondary school. The 0.82% of students in this cluster had more than 10% unauthorised absences in year 8 and more than 60% unauthorised absences in year 11.

## Absence trajectories and achievement

After adjusting for school absenteeism risk factors, Fig 2 depicts the differences in achievement by absence trajectory compared to the Consistently Low Absence (CLA) trajectory. For our binary outcome of five or more GCSE passes, differences are presented as percentage points. In contrast, effect sizes for continuous outcomes (Attainment 8 score, English GSCE score, and Math GCSE score) are presented as standard deviations. Bivariate statistics are presented in the S9 File, while the complete regression tables are presented in the S10 File.

Even after controlling for all risk factors for school absences, large differences in student achievement persist across absence trajectories. For all four achievement outcomes, students with moderate or increasing absence trajectories perform worse than those with the *Consistently Low Absence* trajectory. The negative impact on achievement increases from *Consistently*

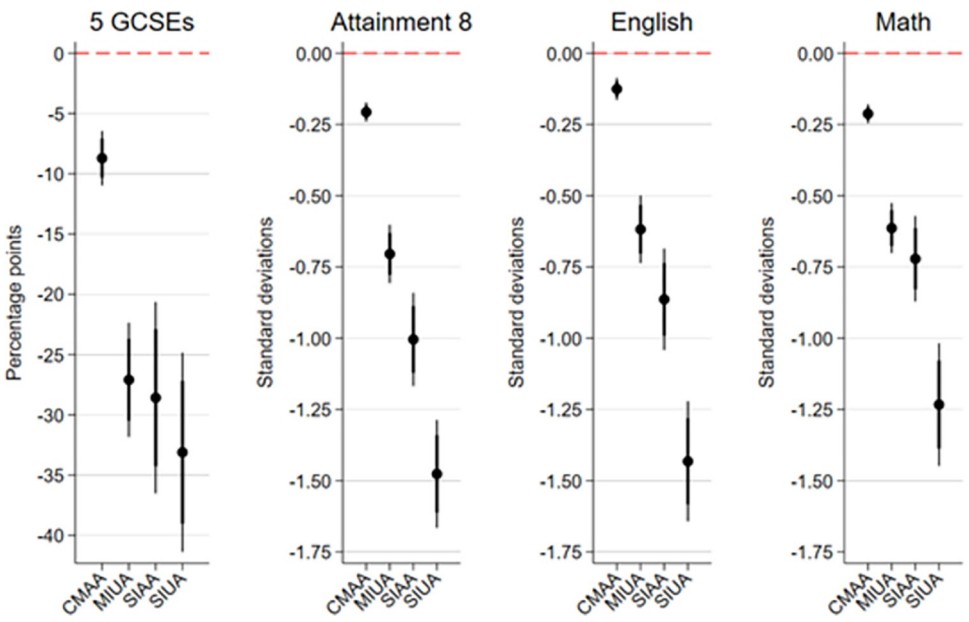

**Fig 2. Differences in achievement by absence trajectory–adjusted for baseline and time-varying risk factors.** Note. Linked MCS-NPD data, N = 7,218, weighted. Reference category: Consistently Low Absence (CLA). CMAA = Consistently Moderate Authorised Absences, MIUA = Moderately Increasing Unauthorised Absences, SIAA = Strongly Increasing Authorised Absences, SIUA = Strongly Increasing Unauthorised Absences. Thick vertical lines indicate the 84%-Confidence Interval, and thin vertical lines the 95%-Confidence Interval.

*Moderate Authorised Absences*, *Moderately Increasing Unauthorised Absences*, *Strongly Increasing Authorised Absences* to its most severe form, *Strongly Increasing Unauthorised Absence*s.

A *Strongly Increasing Unauthorised Absence (SIUA)* trajectory decreases students' probability of obtaining 5 GCSEs by 33.1 percentage points (95%-Confidence interval [CI]: 24.9–41.3). For students with a *Strongly Increasing Authorised Absence (SIAA)* trajectory, the likelihood is lowered to 28.6 percentage points (95%-CI: 20.8–36.4); for students with a *Moderately Increasing Unauthorised Absence (MIUA)*, it is 27.1 percentage points (95%-CI: 22.4–31.8); and for students with a *Consistently Moderate Authorised Absence (CMAA)* trajectory it is 8.7 percentage points (95%-CI: 6.3–11.1). All effects are statistically significant at the 5% level. However, the effect of SIUA, SIAA, and MIUA on obtaining five or more GCSEs was similar (as indicated by the 84%-confidence intervals).

When examining the effects of trajectories on Attainment 8 outcomes, students with a *Strongly Increasing Unauthorised Absence (SIUA)* trajectory have an Attainment 8 score that is 1.48 standard deviations (95%-CI: 1.29–1.67) lower than students with a *Consistently Low Absence Trajectory (CLA)*. A *Strongly Increasing Authorised Absence (SIAA)* trajectory leads to an Attainment 8 score of 1.00 standard deviations lower (95%-CI: 0.84–1.17). The lower attainment among students with a *Moderately Increasing Unauthorised Absence (MIUA)* amounts to 0.70 standard deviations (95%-CI: 0.60–0.81). In comparison, it is 0.21 standard deviations (95%-CI: 0.17–0.24) for students with a *Consistently Moderate Authorised Absence (CMAA)* trajectory. All effects are statistically significant at the 5% level. The effects of each trajectory on Attainment 8 scores were significantly different from one another (as all 84%-confidence intervals do not overlap). The findings for GCSE English and GCSE Math mostly mirror the Attainment 8 results, albeit the difference between SIAA and MIUA in English and Math scores is not statistically significant.

## Discussion

The dynamic nature of absences throughout children's schooling has not been consistently addressed in the prior literature on school absenteeism. Predominantly in the early stages of children's education, numerous studies have examined absences within a single school year or over a brief period. Moreover, the distinction between authorised and unauthorised absence reasons has not been modelled dynamically in prior studies. As unauthorised absences become more prevalent throughout children's schooling, absence trajectories cannot be evaluated in isolation from the type of absence. Our study contributes to the literature by identifying joint authorised and unauthorised absence trajectories throughout a child's compulsory school career for a representative sample of children born in 2000/2001 in England and examining their consequences for academic achievement at the end of compulsory secondary schooling.

Using k-means longitudinal clustering, we found five distinct absence trajectories, each characterised by different levels and patterns of authorised and unauthorised absences. Two-thirds of students fell into a cluster characterised by *Consistently Low Absences*. More than a quarter of students fell into a cluster that exhibited *Consistently Moderate Authorised Absences* from years 1 to 11. Additionally, we found three smaller clusters with increasing levels of authorised and unauthorised absences: *Moderately Increasing Unauthorised Absences*, *Strongly Increasing Unauthorised Absences*, and *Strongly Increasing Authorised Absences*. In contrast to previous research [19, 20, 26], we did not identify a cluster of significantly declining absences over time. However, these previous studies did not investigate absence trajectories over primary and secondary school careers. In addition, unlike previous studies, we examined the joint trajectory of authorised and unauthorised absences.

Furthermore, our study investigated the consequences of these absence trajectories for achievement, accounting for various risk factors associated with school absenteeism. Our findings revealed significant differences in student achievement across absence trajectories. Compared to students with a *Consistently Low Absence* trajectory, students with a *Consistently Moderate Authorised Absence* trajectory face a significant achievement disadvantage of 0.13–0.21 standard deviations, and students in the three increasing absence trajectories face a huge achievement disadvantage of 0.61–1.48 standard deviations. However, there is also substantial heterogeneity in student achievement in the three increasing trajectories, with students in the *Moderately Increasing Unauthorised Absence* trajectory being least disadvantaged and students with a *Strongly Increasing Unauthorised Absence* trajectory being most disadvantaged.

Our findings show that both timing of absences and cumulative exposure matter. Consistent attendance over multiple years ensures students receive the necessary instruction and support to master new concepts and skills. In addition, it provides opportunities for students to develop long-lasting positive peer networks and relationships with teachers. Any transition to more frequent absences, whether unauthorised or authorised, derails these gains, and has long-term consequences for academic performance. The vast differences in achievement between pupils in different absence trajectories underscore the importance of modelling absences in a holistic and dynamic way. Relying on snapshot measures or limited periods underestimates the severity of the detrimental consequences of absences for achievement. The findings, therefore, emphasise the importance of examining entire trajectories of absenteeism and its associations with academic outcomes. They further stress the importance of distinguishing the reason for absence because trajectories of unauthorised vs. authorised absences have different negative effects on achievement.

The findings also question the predominant focus on "chronic" or "persistent" absenteeism, often defined as students missing more than 10% of classes in a certain period. Neither students in the *Consistently Low Absence* trajectory nor students in the *Consistently Moderate*

*Authorised Absence* trajectory break the threshold of 10% absences per year. However, over the 11 years, their absences accumulate and substantially impact achievement. Considering that students could have attended about 1,700 schooldays, students in the *Consistently Moderate Authorised Absence* trajectory missed about 80 schooldays more than students in the *Consistently Low Absence* trajectory. These important differences are missed when focusing on only chronic absenteeism. Likewise, students in all three increasing absence trajectories have, on average, absences above 10% per year throughout secondary school but differ drastically in achievement.

Our study has significant limitations. First, this study describes the absence trajectories of students born in 2000 and 2001 from 2006 to 2017. The absence trajectories of more recent cohorts of students affected by school closures during the Covid-19 pandemic may differ and have a different impact on academic achievement. Second, because information on private schools is not included in the NPD data, our findings cannot be generalised to the entire population of pupils in England. Third, while we have accounted for many significant confounders of the relationship between absence trajectories and achievement, we cannot rule out the possibility of unobserved heterogeneity. As a result, we must proceed with caution when drawing causal conclusions from our findings. Finally, absence trajectories and their impact on achievement may differ by sociodemographic groups, such as gender. Future research should investigate this possibility, particularly if population-level data is available to ensure sufficient statistical power.

Despite these limitations, our research has significant educational policy and practice implications. Since schools reopened following the Covid-19 pandemic, there has been an alarming increase in school absenteeism rates in the UK and globally. For example, England's absence rate rose from 4.3% in the autumn of 2018 to 7.5% in 2022 [3]. While our findings apply to a pre-pandemic cohort, identifying typical absence trajectories in this study lays the groundwork for understanding how specific absenteeism patterns may evolve in cohorts after the pandemic. Furthermore, it emphasises the likely extent of the negative consequences of moderate and escalating absence trajectories, whether authorised or unauthorised, on future achievement in a post-pandemic world if not addressed by policy and practice.

The current emphasis on minimum attendance thresholds and penalties for non-compliance in policy frameworks may not effectively address the diverse absence trajectories that our study has uncovered. Various absence trajectories from consistently low to strongly increasing absences suggest that a more nuanced and individualised approach to addressing absenteeism post-pandemic is necessary. Tailored interventions are required depending on specific patterns of absences. For instance, while moderate levels of absences might require supporting and encouraging attendance, patterns of increasing absences will require interventions that address root causes, such as underlying health conditions. Furthermore, implementing policy measures for the early detection of attendance issues is critical, allowing for timely intervention and support to prevent further disruptions to students' education after the pandemic. Collaboration among schools, policymakers, and educators is important to proactively identify and help at-risk students, with the goal of preventing any decline in their academic progress.

Our findings also highlight the importance of developing the capacity of schools to analyse and identify school attendance patterns. This will enable them to implement targeted interventions to address the needs of those with specific patterns of absences. For instance, practitioners will benefit from considering the specific reasons behind different patterns of absenteeism and developing flexible and responsive policies that consider each student's unique circumstances. Finally, the findings underscore the importance of a proactive and preventive approach to absenteeism rather than a reactive one. Educators, parents, and policymakers must collaborate to create support systems and resources to help students overcome challenges

before absenteeism becomes a significant issue. This will involve identifying root causes of absenteeism, such as health issues, family circumstances, or academic challenges, and developing targeted interventions to address these underlying issues.

In conclusion, our research on school absenteeism underscores the need for a nuanced and dynamic approach to educational policy and practice. By identifying diverse absence trajectories throughout a child's compulsory school career and revealing significant consequences for academic achievement, we emphasise the importance of examining patterns of attendance rather than the predominant focus on attendance thresholds and penalties. Our findings highlight the importance of early intervention and developing the capacity of schools to analyse and identify patterns of absences. It also stresses the need for a collaborative approach among educators, parents, and policymakers to address the varied reasons behind absenteeism and develop tailored interventions based on each student's unique circumstances to provide a supportive and responsive educational environment for all learners.

## Supporting information

**S1 File. Sample selection.**
(DOCX)

**S2 File. Inverse probability weights.**
(DOCX)

**S3 File. Distribution of absences.**
(DOCX)

**S4 File. Timing of measurement of covariates.**
(DOCX)

**S5 File. Measurement of latent factors.**
(DOCX)

**S6 File. Distribution of achievement and covariates.**
(DOCX)

**S7 File. Number of clusters.**
(DOCX)

**S8 File. Yearly mean absences in preferred cluster solution.**
(DOCX)

**S9 File. Bivariate associations between absence trajectories and achievement.**
(DOCX)

**S10 File. Full regression tables.**
(DOCX)

## Acknowledgments

We are grateful to the advisory group of our Nuffield Foundation project (FR-000023241) for their comments on an earlier version of this manuscript as well as to the anonymous reviewers. We are also grateful to the Centre for Longitudinal Studies (CLS), UCL Social Research Institute, for the use of these data and to the UK Data Service for making them available. However, neither CLS nor the UK Data Service bear any responsibility for the analysis or interpretation of these data.

## Author Contributions

**Conceptualization:** Jascha Dräger, Markus Klein, Edward M. Sosu.

**Data curation:** Jascha Dräger.

**Formal analysis:** Jascha Dräger, Markus Klein, Edward M. Sosu.

**Funding acquisition:** Markus Klein, Edward M. Sosu.

**Writing – original draft:** Jascha Dräger, Markus Klein, Edward M. Sosu.

**Writing – review & editing:** Jascha Dräger, Markus Klein, Edward M. Sosu.

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
