## [Decision Letter · Decision Letter 0]

13 Mar 2024

PONE-D-24-01059Trajectories of School Absences Across Compulsory Schooling and Their Impact on Children's Academic Achievement: An Analysis based on linked longitudinal survey and school administrative dataPLOS ONE

Dear Dr. Klein,

Thank you for submitting your manuscript to PLOS ONE. After careful consideration, we feel that it has merit but does not fully meet PLOS ONE’s publication criteria as it currently stands. Therefore, we invite you to submit a revised version of the manuscript that addresses the points raised during the review process.

We look forward to receiving your revised manuscript.

Kind regards,

Cathryn Knight

Academic Editor

PLOS ONE

Journal Requirements:

"The project has been funded by the Nuffield Foundation [grant number FR-000023241]."

**Additional Editor Comments:**

Thank you for submitting this paper. It offers a novel insight into the impact of absences on education in England. In addition to the reviewers thought below, I have a few suggestions of minor revisions before I believe this paper is suitable for publication.

- A discussion on policy around absences in England at the time of data collection would be useful - this could be compared with the current policy in the discussion in relation to what you found. 

- A consort diagram in the methods would be useful to understand who was excluded from analysis. For example, were there learners who had missed years of schooling or who did not have GCSEs?

- I limitations section which acknowledges that the data is now approx 8 years out of date (since MCS cohort did GCSEs) would be useful. 

Reviewers' comments:

Reviewer's Responses to Questions

**Comments to the Author**

1. Is the manuscript technically sound, and do the data support the conclusions?

Reviewer #1: Yes

2. Has the statistical analysis been performed appropriately and rigorously? 

Reviewer #1: Yes

3. Have the authors made all data underlying the findings in their manuscript fully available?

Reviewer #1: No

4. Is the manuscript presented in an intelligible fashion and written in standard English?

Reviewer #1: Yes

5. Review Comments to the Author

Reviewer #1: Thank you for allowing me to review this excellent paper. I really enjoyed the read. On a general note this is a very well written and clear paper, which examines the important issue of school absence, which we know has increased massively since the pandemic, so this is a very timely topic.

Further comments are below addressing each specific sections of the paper. There are some minor suggestions for improvement.

INTRODUCTION

The introduction sets out the previous literature and the justification for undertaking the current study extremely well.

METHODS

I am impressed with the careful use of weights, as well as imputation of missing data, in order to get as close a possible to nationally representative estimates for this population. Also the control variables are extremely rich and all very relevant. The use of advanced methods to identify clusters is very commendable.

RESULTS

The results are very well presented. I really like the graphs which make it easy to follow. One minor comment is that you may want to put 95% confidence intervals the SD results in the text itself for good measure.

I am super curious to know whether you found and gender differences in results. Maybe the nice graph in figure two could be done by gender and included as an Appendix and in the text you could refer these as additional analyses and comment on whether girls an boys are similarly affected.

DISCUSSION

If space permits, if would be good to highlight the huge increase in absence from before the pandemic: https://www.gov.uk/government/statistics/pupil-absence-in-schools-in-england-2018-to-2019 compared to after the pandemic: https://explore-education-statistics.service.gov.uk/find-statistics/pupil-absence-in-schools-in-england

This would highlight and support the importance of your findings for the current generation of school children who miss school.

Although the paper and the study is methodologically very sound, it would be good to see you list some study limitations. E.g. that although significant efforts have been made to control for confounding factors, because this is an observational study, care has to be taken with interpreting results a causal.

OTHER

I know that the data cannot be published due UK Data Service licence restrictions, and you have stated this. However, I suggest that you make your analytical code available to readers on GitHub for example. The research community could really benefit from this when conducting a similar study.

6. PLOS authors have the option to publish the peer review history of their article (what does this mean?). If published, this will include your full peer review and any attached files.

Reviewer #1: No

---

## [Author Response · Author response to Decision Letter 0]

21 Apr 2024

A "Response to Reviewers" document was submitted.

---

## [Editor Report · Decision Letter 1]

23 Jun 2024

Trajectories of School Absences Across Compulsory Schooling and Their Impact on Children's Academic Achievement: An Analysis based on linked longitudinal survey and school administrative data

PONE-D-24-01059R1

Dear Dr. Klein,

We’re pleased to inform you that your manuscript has been judged scientifically suitable for publication and will be formally accepted for publication once it meets all outstanding technical requirements.

Kind regards,

Leonard Moulin

Academic Editor

PLOS ONE

Additional Editor Comments (optional):

Thank you for addressing the reviewer's and editor's comments and for explaining the improvements to the manuscript. The review of the revised version of your manuscript has taken an unusually long time due to the change of editor.
---

## [Editor Report · Acceptance letter]

2 Jul 2024

PONE-D-24-01059R1 

PLOS ONE

Dear Dr. Klein, 

I'm pleased to inform you that your manuscript has been deemed suitable for publication in PLOS ONE. Congratulations! Your manuscript is now being handed over to our production team.

Kind regards, 

on behalf of

Dr. Leonard Moulin 

Academic Editor

PLOS ONE